# Microseismic Dynamic Response and Multi-Source Warning during Rockburst Monitoring Based on Weight Decision Analysis

**DOI:** 10.3390/ijerph192315698

**Published:** 2022-11-25

**Authors:** Jiawei Tian, Dong Chen, Zhentang Liu, Weichen Sun

**Affiliations:** 1School of Safety Engineering, China University of Mining and Technology, Xuzhou 221116, China; 2State Key Laboratory for Geomechanics & Deep Underground Engineering, China University of Mining and Technology, Xuzhou 221116, China

**Keywords:** microseismic, rockburst, source energy, time-frequency characteristics, monitoring, warning

## Abstract

**Highlights:**

**What are the main findings?**
Fourier transform, wavelet packet transform and Hilbert-Huang transform are used to analyze microseismic events, and the time-frequency characteristics of high-energy events are obtained.The unascertained measure comprehensive evaluation model based on 1366 working face of Hengda coal mine is established.

**What is the implication of the main finding?**
It provides a method for the deep analysis of microseismic signals and provides a basis for the risk assessment of rock burst.The establishment of the model based on weight decision analysis makes the multi-parameter monitoring of rock burst more efficient and accurate.

**Abstract:**

To prevent rockburst disasters and improve the accuracy of warnings for rockburst, based on the microseismic data of the 1366 working face of Hengda Coal Mine collected by the microseismic monitoring system, Fourier transform, wavelet packet transform, and Hilbert–Huang transform analysis methods are used for time-frequency domain joint analysis. The time-frequency differences of the main frequency, amplitude, frequency band percentage, and instantaneous energy of the high-energy microseismic event and the events before high-energy microseismic event are obtained. The analysis shows that the high-energy event has obvious low frequency characteristics, and when the high-energy event occurs, the instantaneous energy shows an obvious “inverted V” trend. At the same time, it is found that the acoustoelectric indexes show a trend of “rising” or “inverted V” when the high-energy event occurs. On this basis, the unascertained measure comprehensive evaluation model of rock burst hazard is established by analytic hierarchy process (AHP). Based on the analysis of microseismic data and the acoustoelectric index of the 1366 working face in Hengda coal mine, it is of great significance to determine the warning indicators for rockburst, improve the accuracy of uncertainty quantitative analysis for rockburst, and improve the discrimination accuracy of rockburst risk.

## 1. Introduction

With the gradual increase of coal mining depth in China, coal mining has also gradually entered the deep mining stage, and the risks caused by coal and rock dynamic disasters are more and more worthy of attention, and analysis of coal and rock dynamic disasters is of great significance [1,2,3]. Among them, rockburst is a dynamic phenomenon characterized by sudden, sharp, and violent damage caused by the instantaneous release of elastic properties of coal and rock masses around coal mines, roadways, or working faces [4,5], which is often accompanied by instantaneous displacement, throwing out, loud noise and air waves of coal and rock masses. In serious cases, it will cause casualties and damage to roadways, and even cause surface collapse and local earthquake [6]. Rock burst seriously threatens the safe and efficient production of coal mines and can cause serious casualties and economic losses [7,8,9]. Therefore, strengthening the monitoring and early warning of rock burst will effectively reduce the disaster caused by rock burst and ensure the safety of mine production.

At present, there are many means for warning of rockburst, such as mining stress monitoring [10], microseismic monitoring, ground sound monitoring, acoustic emission monitoring [11,12,13], electromagnetic radiation monitoring, etc. Many scholars have studied different warning methods. Wang Enyuan et al. [14] proposed the collaborative warning technology of acoustic emission and electromagnetic radiation for coal and rock dynamic disasters, realizing the organic integration and complementary advantages of electromagnetic radiation and acoustic emission technology; Tian Xianghui et al. [15] proposed a quantitative trend rockburst hazard warning method based on daily maximum microseismic energy and total number of high values of microseismic energy/frequency deviation. He Xueqiu, Dou Linming et al. [16] used microseismic frequency index and energy index to monitor the mining disturbance and geological anomaly of coal seam, proposed a new method for regional dynamic monitoring of microseismic in outburst dangerous coal seam, studied the theoretical basis for regional monitoring by microseismic monitoring technology, and established the corresponding index evaluation system. Xu wenquan, Wang Enyuan et al. [17] developed a mining stress monitoring system for coal and rock mass, which can be used for warning of coal and rock dynamic disasters, such as rock burst [18].

Among these monitoring and early warning methods, microseismic monitoring technology, as a real-time, dynamic, and continuous monitoring method, plays an increasingly prominent role in the monitoring and early warning of rock burst. The microseismic monitoring technology can monitor the vibration generated by the fracture of coal and rock mass [19,20], evaluate the damage status of coal and rock mass, and has the characteristics of real-time, dynamic, and continuous monitoring [21]. It can ascertain the location, energy, frequency, and other information of microseismic events in real time and accurately, then judge the stability of rock structure, so as to provide warning information for coal and rock dynamic disasters such as rock burst [22,23,24]. It has been widely used in the warning practice of underground engineering rockburst in different fields [25,26,27]. In addition, the dynamic response and characteristic analysis of microseismic signals in the process of microseismic monitoring is the key to improve the accuracy of microseismic monitoring [28,29,30]. Therefore, this paper conducts an in-depth analysis of the microseismic monitoring data based on waveform signals.

Although many scholars have done a lot of research on rockburst warning, there are few studies on the depth analysis of microseismic monitoring information and the fusion of multi-source monitoring indicators based on microseismic monitoring information, so it is hard to accurately determine the weight of each indicator in multi-parameter monitoring. On the basis of previous research, this paper applies three time-frequency analysis methods of Fourier transform, wavelet packet transform, and Hilbert–Huang transform to extract the characteristics of microseismic signals. Then, the microseismic monitoring combined with acoustoelectric monitoring is used for multi-source warning, and the analytic hierarchy process (AHP) is used to determine the weight of each index. Finally, the unascertained measure comprehensive evaluation model in the 1366 working face of Hengda coal mine is established. In this way, the monitoring and warning method based on weight analysis can highlight the importance of each index. Through quantitative analysis, the influence proportion of each index can be obtained, and then multi parameter fusion can improve the accuracy of monitoring and warning. By means of the depth analysis of the signal in the time-frequency domain, the method of weight decision analysis combined with other multi-parameters can also be used in working faces of other coal mines. It can also provide a method for multi-parameter monitoring and early warning of rock burst, improve the accuracy of rock burst monitoring and early warning, and ensure the safety of mine production.

## 2. Site Conditions

### 2.1. Working Face Position

The 1366 working face is located in the 126 mining area of Hengda Coal Mine. The length of the haulage roadway is 590 m and the mining length is 487 m. The length of air-return roadway is 485 m, the mining length is 485 m, and the opening hole length is 142~186 m; The mining area is 83,575 square meters. The mining elevation of this working face is −654.5~−691.1 m, the surface elevation is 169.5~178 m, and the vertical depth with the working face is 831.7~869.1 m. The 1366 working face is located at the axis of Wangying syncline. The coal seam has a wide and gentle syncline structure. The coal seam strikes N 48–70°, the south coal seam inclines to southeast, the north coal seam inclines to northwest, and the coal seam dip angle is 1~7°.

The 1366 working face was identified as having moderate rockburst tendency, with many microseismic events with energy above 10^4^ J and frequent occurrence of coal-rock dynamic phenomena with large energy.

### 2.2. Conditions of Coal Seam

The mining coal seam of 1366 working face is water layer 2 and water layer 3 of water layer group. The thickness of water layer 2 is between 1.6 m and 2.4 m, with an average thickness of 1.94 m. The structure of coal seam is simple, and there are 1~2 layers stone, with a thickness of 0.1~0.2 m. The thickness of water layer 3 is between 2.8 m and 3.9 m, with an average thickness of 3.43 m. The structure of the coal seam is simple. The thickness of 1~2 layers is between 0.1 m and 0.3 m. Water layer 2 and water layer 3 coal seam group stone thickness is 0.2~1.6 m, from north to south thinning.

### 2.3. Layout of Microseismic Monitoring System in Working Face

SOS microseismic monitoring system can carry out remote, real-time, dynamic, and automatic monitoring of mine microseismic signals, and record the complete waveform of microseismic events. Through data processing and analysis, the time, energy, and spatial three-dimensional coordinates of vibration with energy greater than 100 J and frequency between 0 Hz and 600 Hz can be monitored, the vibration type can be judged, the force source of vibration can be determined, and the real-time evaluation of the risk degree of dynamic phenomena in the monitoring area can be realized.

The SOS microseismic monitoring system is used to monitor the production activity range of the 1366 working face. According to the recorded data, the occurrence trend of rock burst hazard can be evaluated, which provides a basis for evaluating the rock burst hazard within the production activity range of the 1366 working face. The system is equipped with 4 microseismic sensors on the 1366 working face. The arrangement of microseismic sensors on the 1366 working face is shown in Figure 1.

## 3. Time-Frequency Characteristics of Microseismic Signals in Working Face

### 3.1. Time Series Evolution of Energy of Microseismic Event in January in Working Face

The microseismic events that occurred at the 1366 working face of Hengda Coal Mine are analyzed, and the sequential distribution of microseismic energy within the monitoring range is shown in Figure 2.

From the sequential evolution statistical chart of microseismic energy in January, it can be seen that the energy distribution of most microseismic events in the 1366 working face in January is below 10^4^ J, the magnitude of maximum and total energy of microseismic events on only some dates exceeded 10^4^ J, and microseismic events with energy greater than 10^4^ J are considered as events with large rockburst hazard degree and impact range. As can be seen from Figure 2, in the microseismic event on January 6, the maximum energy of a single event reached 5 × 10^4^ J, reaching the highest value in January. The event is considered a high-energy event. Such events can easily lead to coal and rock dynamic disasters such as rockburst. Therefore, the high-energy event on January 6 and the events before high-energy event are taken as the research objects for analysis.

The basic information of microseismic events studied in this paper is shown in Table 1. 

In Table 1,Microseismic event 1 is a coal-rock dynamic event, i.e., a high-energy event.

The spatial distribution diagram and energy cloud diagram of microseismic events on the 1366 working surface listed in Table 1 are shown in Figure 3 and Figure 4 (red event is high-energy event, and other blue events are the events before the high-energy event).

### 3.2. Spectrum Analysis Based on Fourier Transform

The data collected by the microseismic sensors can only directly provide the information in the time domain, but some information that cannot be seen in the frequency domain can be found by using the spectrum analysis. Using Fourier transform for signal, the stable signal can be completely analyzed in the frequency domain, and the frequency composition, signal energy, and signal distribution in the frequency domain can be obtained.

Through Fourier transform of the microseismic signal, the spectrum of high-energy event and events before the high-energy event are obtained, as shown in Figure 5.

By comparing the high-energy event with other events, the following characteristics can be found:

The amplitude of the dominant frequency of the high-energy event is large, and it is high in the frequency range of 0–50 Hz, with the order of magnitude reaching 10^−5^. The vibration energy level of the whole event is large, reaching 10^4^ J, and the impact hazard degree and influence range are large. However, the amplitude of each microseismic event before the occurrence of high-energy event is less than 10^−5^, and the vibration energy level of the event is small, far less than 10^4^ J, resulting in a small impact harm degree and impact range. This indicates that before the occurrence of this high-energy event, there is a period of low energy release, which is called the energy accumulation period. After the energy accumulation period, a large amount of energy is released in a short time, which is manifested as a high-energy event.

It can be seen from Figure 4 that the dominant frequencies of the spectrum of the high-energy event and the events before the high-energy event are 5.53 Hz, 198.57 Hz, 189.94 Hz 98.95 Hz, 198.24 Hz, 198.57 Hz, 198.57 Hz, 96.78 Hz, 98.31 Hz, and 98.47 Hz. It can be seen that the dominant frequency at the occurrence of the high-energy event is within 10 Hz, and the dominant frequency of the events before the occurrence of high-energy event is mostly greater than 100 Hz. The low frequency components of the high-energy event account for a large proportion, especially those below 100 Hz, while those below 100 Hz account for a small proportion in the previous events.

Through the comparative analysis of the spectrum signal characteristics of different events, it is shown that the high-energy event is a low frequency event with high amplitude, indicating that they carry high energy, which is also the reason for the occurrence of coal-rock dynamic phenomena such as rock burst.

### 3.3. Frequency Band Energy Analysis Based on Wavelet Packet Transform

Fourier Transform is suitable for the analysis of global features, but it is not good for mutation signals and nonstationary signals, and it is deficient in time domain analysis. Therefore, wavelet packet transform can be used to supplement the defects of Fourier Transform.

Wavelet packet transform is a local transformation of time and frequency, which can simultaneously conduct multi-scale joint analysis of data in the time and frequency domain. It has the function of multi-scale refinement analysis, which is suitable for the analysis of local characteristics of signals and has a better effect on nonstationary signals.

The wavelet packet transform formula is as follows:(1)$$\mathrm{WT}\left(a,\tau \right)=\frac{1}{\sqrt{a}}\int_{-\infty }^{\infty }f\left(t\right)*\psi \left(\frac{t-\tau }{a}\right)\mathrm{dt}$$
where WT is wavelet transform, t is time, a is the scale and τ is the translation.

Wavelet packet transform is used to analyze the frequency band energy of the high-energy event and the events before the occurrence of high-energy event. The db8 wavelet basis function is selected to decompose the microseismic signal into 5 wavelet packets. In the fifth layer, there are 32 wavelet packets, and the whole frequency domain is divided into 32 sub-bands. The percentage of the energy of each node in the fifth layer wavelet packet decomposition is calculated to the total energy of microseismic signal. The results are shown in Figure 6.

It can be seen that the highest energy distribution frequency band of the high-energy event and the events before the occurrence of high-energy event is 1, 25, 15, 25, 11, 8, 10, 7, 25, and 9, respectively. When a high-energy event occurs, the highest energy distribution frequency band of the event is within 5, while before the high-energy event occurs, the highest energy distribution frequency bands of the events are mainly concentrated in the frequency band of 11~32.

The energy distribution of event 1 is concentrated in frequency bands 1–10, in which the energy distribution of frequency bands 1–5 accounts for 62.50% of the total energy, and the energy distribution of frequency bands 1–10 accounts for 83.54% of the total energy. This indicates that event 1 is a low frequency event.

The energy distribution of microseismic events 2–10 is concentrated in the 11–32 frequency band, and the energy distribution in the 11–32 frequency band accounts for 70.08%, 75.81%, 67.91%, 67.78%, 34.46%, 65.89%, 46.52%, 62.21%, and 63.81% of the total energy, most of which are higher than 60%. Compared with event 1, the frequency distribution of events 2–10 is more dispersed, and most of them are concentrated in the medium and high frequency bands.

It can be seen that the frequency band energy distribution of microseismic signals is different before and during the occurrence of the high-energy event. It has obvious low-frequency characteristics when the high-energy event occurs, and the release of energy is relatively greater.

### 3.4. Hilbert Huang Transform (HHT)

Fourier transform and wavelet packet transform cannot achieve high accuracy in both time and frequency. However, the Hilbert–Huang transform can overcome this defect and achieve high accuracy in both time and frequency, which makes it very suitable for analyzing mutational signals.

HHT is a time-frequency analysis method, which gets rid of the constraints of linearity and stability. It can achieve high accuracy both in time and frequency. The main content of HHT consists of two parts. The first part is empirical mode decomposition (EMD), which was proposed by Huang. The second part is Hilbert spectrum analysis.

In short, the basic process of HHT processing nonstationary signals is: Firstly, the EMD method is used to decompose the given signal into several intrinsic mode functions, which are expressed by IMF, and these IMF are components that meet certain conditions. Then, Hilbert transform is performed on each IMF to obtain the corresponding Hilbert spectrum, that is, each IMF is represented in the joint time-frequency domain. Finally, the Hilbert spectrum of the original signal can be obtained by summarizing the Hilbert spectrum of all IMF.

The waveform of microseismic events is analyzed by the HHT method, and 12 different IMF components are obtained through EMD, as shown in Figure 7.

Comparing event 1 with the other nine microseismic events, the maximum amplitudes of all subgraphs of event 1 are obviously higher than the other nine events as a whole. For example, in the IMF component subgraph decomposed by event 1, the amplitude of the first subgraph is 7.04 × 10^−4^. However, in the first IMF component subgraph of other events, the magnitude of the maximum amplitude is mainly concentrated in 10^−5^ and 10^−6^.

In addition to the difference in maximum amplitude in the subgraph, the IMF components of each microseismic event were compared in the EMD plots, it can be seen that among the IMF components from 1 to 7, the waveform in event 1 is relatively gentle with small amplitude in the early sampling period, and the amplitude increases between 1800 and 4000 sampling points in the middle period, and obvious waveform appears. After 4000 sampling points, the waveform flattens out again. However, for the 8–12 IMF components, the complete waveform start–end period was not recorded in these components during the sampling time, and the difference was relatively rare in the sampling time due to the low frequency, so this paper will not discuss it.

In comparison with the other nine microseismic events before the occurrence of high-energy event, the variation of the high frequency component IMF1–7 is obviously different from that shown in event 1. Firstly, the waveform duration of each component after EMD of events 2–10 is relatively long, and most of the waveform duration lasts for the whole sampling time. Moreover, the characteristics of event 1 are the opposite of events 2–10. Secondly, the frequencies of the components of events 2–10 also increase. This indicates that the energy release of the high-energy microseismic event is more concentrated than that of low-energy microseismic events before the high-energy event occurrences, and more energy can be released in a short period.

When the high-energy event occurs, the frequency decreases, which indicates that when the high-energy event or even rockburst occurs, the energy release is in a lower frequency state. At this time, the wavelength is longer, which leads to the longer propagation distance and larger influence range.

Using HHT to analyze the instantaneous energy of each microseismic event waveform, the results are shown in Figure 8.

As can be seen from Figure 8, the maximum instantaneous energy of the waveform of event 1–10 is 2.16, 1.15, 1.26, 1.30, 1.60, 1.79, 1.50, 1.65, 1.69, and 1.63, respectively. Only when event 1 occurred, the maximum instantaneous energy of microseismic reached 2.16. However, the highest instantaneous energy of events 2–10 is less than 2. This phenomenon shows that the high-energy event releases higher energy in a short time interval, which corresponds to the occurrence mechanism of rockburst. In addition, the instantaneous energy of the waveform of events 2–10 is not high, indicating that there is a quiet period of energy before the occurrence of rockburst.

Further analysis of the instantaneous energy diagram of microseismic events shows that, among the 0–6000 sampling points sampled by sensors, the high value of instantaneous energy of event 1 is mainly distributed between 1800 and 2800 sampling points, and the instantaneous energy tends to 0 in the rest of the time interval. It shows a trend that rises steeply from zero to the highest point and then drops steeply to zero, namely the “inverted V type” trend, which is reflected in the instantaneous release of higher energy. The overall instantaneous energy of microseismic events 2, 3, 4, 5, 7, 9 and 10 is higher in all 0–6000 sampling points, most of which are above 0.2, and compared with microseismic event 1, instantaneous energy does not show a single “inverted V type” trend that rises steeply from zero to the highest point and then drops steeply to zero, but shows a fluctuation pattern that fluctuates in the whole sampling interval. Although event 6 and event 8 also show an “inverted V type” trend similar to event 1, compared with event 1, except for the peak value within the sampling time, the instantaneous energy of other sampling points does not approach 0, but fluctuates up and down at a fixed value.

## 4. Analysis of Acoustoelectric Precursor Rules

By analyzing the waveform of microseismic events, we can know the precursory rules of coal-rock dynamic events. However, if there is only a single monitoring method, the monitoring of high-energy coal-rock dynamic events will inevitably lead to misjudgment or low monitoring accuracy. Therefore, acoustic emission (AE) and electromagnetic radiation (EMR) can be introduced for multi-parameter comprehensive analysis to improve the monitoring accuracy of coal-rock dynamic phenomena.

Statistics of acoustic emissions and electromagnetic radiation intensity for the occurrence date and extension date of the microseismic events analyzed above are shown in Figure 9.

From the acoustoelectric statistical chart from December 27 to January 14, it can be seen that the change trend of acoustic emissions and electromagnetic radiation intensity is basically the same. The acoustic emissions present an “inverted V type”, and during this period, the intensity of acoustic emissions increased continuously and reached the peak on 4 January. Later, in the process of decline, the coal-rock dynamic event occurred on 6 January, and then the acoustic emissions intensity continued to decline. The electromagnetic radiation presents a “rising type”. The intensity of electromagnetic radiation continued to rise before 6 January, still increased after the high-energy event on January 6, and began to decline after 10 January. This indicates that acoustic emissions and electromagnetic radiation have obvious reactions before the occurrence of the high-energy event.

Through the analysis of acoustoelectric indicators before the occurrence of the high-energy event and consulting relevant data, the following criteria for determining the risk of acoustic emissions and electromagnetic radiation can be determined:

Electromagnetic radiation intensity or acoustic emission intensity has an obvious trend of continuous enhancement, namely “rising type”, the trend is close to the critical warning value, and the trend change rate exceeds the trend warning value, and the duration is greater than the minimum duration, $\left|\eta \right|\geq \eta_{C}$ and $T\geq T_{C}$.

The electromagnetic intensity or acoustic emissions intensity in a certain area increases significantly to the maximum value in the near future and then decreases, which is the “inverted V type”. In particular, the most dangerous situation is that it continues to rise to the maximum value in the near future and then drops to the minimum value in the near future.

When it is confirmed that there is no interference, little interference, or no fault, the electromagnetic radiation intensity or acoustic emissions intensity changes strongly, and the variation range exceeds the trend warning value.

For the warning value mentioned in the judgment criteria, it should be determined according to the specific geological and mining conditions, mine pressure, and the variation rules of acoustic emissions and electromagnetic radiation.

## 5. Unascertained Measure Comprehensive Evaluation Model for Rockburst Hazard

Uncertainty is the uncertainty in cognition because the evidence people have is not enough to grasp the real state of things. Liu Kaidi et al. proposed an evaluation model, namely the unascertained measure model, which uses a real number between [0, 1] to describe whether things are in an unascertained state or have unascertained properties. Because rockburst is an uncertain dynamic disaster caused by coal and rock properties, mining influence, and other factors, which cannot be completely predicted accurately, the unascertained measure model can be used to evaluate the risk of rockburst.

Through the analysis of microseismic monitoring and acoustoelectric monitoring of the 1366 working face in Hengda coal mine, the critical value of rockburst warning index of the 1366 working face in Hengda coal mine can be preliminarily determined:

The dominant frequency of microseismic event waveform is within 0–100 Hz, and the low-frequency component accounts for a large proportion. The amplitude of the dominant frequency exceeds 10^−6^. The frequency band of the highest energy distribution of wavelet packet decomposition of microseismic events is within 10, and most of the energy distribution is concentrated in the frequency band of 0–10.

After the microseismic event is transformed by HHT, the maximum amplitude of the first IMF component obtained exceeds 10^−4^.

The instantaneous energy obtained by HHT exceeds 2, and shows an “inverted V type” trend that rises steeply from zero to its peak and then drops steeply to zero.

Acoustic emissions (AE) and electromagnetic radiation (EMR) show an obvious trend of “rising type” or “inverted V type” or other obvious fluctuations in recent days.

In this paper, each index is divided into three levels by using the grading quantitative method, (Grade I, II, and III, respectively, mean no rockburst risk, medium rockburst risk, and strong rockburst risk) each level specifies a value standard and value (The value is 1, 3, 5). The grading of quantitative indicators is easier than that of qualitative indicators. The scoring standard can be divided into three sections. According to a large number of statistical analyses, numerical simulations, existing standards, and research results, each index can be classified. See Table 2 for details.

The analytic hierarchy process (AHP) is a research method that combines qualitative and quantitative weight calculation, establishes a matrix by comparing various factors, and finally obtains the importance of each judgment indicator. In the evaluation indexes of rockburst tendency, there are not only quantitative indexes, such as microseismic event energy and amplitude of main frequency of microseismic event, but also qualitative indexes, such as acoustoelectric indexes, that cannot be completely quantified. Therefore, the AHP can be used to determine the weight of various indicators of rockburst tendency.

In the monitoring and warning of rockburst, the AHP is used to carry out weight analysis, which can effectively quantify each index. Moreover, after obtaining the importance degree of each index, the possibility of rockburst in coal and rock mass can be further judged, so as to take measures to prevent and control it.

The key part of AHP is how to construct a judgment matrix, so as to compare the importance of different indexes. The construction of the judgment matrix can be determined according to the 1–9 scale criterion in Table 3.

Compare and judge different indicators through the scale established in Table 3. According to relevant data and the sensitivity of different evaluation indexes to coal-rock dynamic events and rockburst events, the following judgment matrix is established.
(2)$$\left[\begin{array}{cccc} a_{11} & a_{12} & \ldots & a_{1j}\\ a_{21} & a_{22} & \ldots & a_{2j}\\ \ldots & \ldots & \ldots & \ldots \\ a_{i1} & a_{i1} & \ldots & a_{ij} \end{array}\right]$$
where $a_{ij}$ indicates the importance of $A_{i}$ compared to $A_{j}$.

According to the established judgment matrix analysis measure model, the weight of each evaluation index is obtained, as shown in Table 4.

By classifying various indexes, such as the microseismic and acoustoelectric index of the working face, and determining the specific weight of each index, an unascertained measure comprehensive evaluation model of the 1366 working face of Hengda coal mine can be established to preliminarily evaluate the rockburst tendency of the 1366 working face.

Finally, the equation of the evaluation system is as follows:(3)$$\omega =\frac{\sum_{i=1}^{n}C\times \alpha_{i}}{5}$$
where ω is the rockburst tendency at a certain moment; C is the grading index of rockburst tendency evaluation index, with values of 1, 3, and 5; n is the number of evaluation indexes, and $\alpha_{i}$ is index-weight of the indicators.

Through the verification of microseismic events and coal-rock dynamic phenomena occurring at different times, it can be set that if ω exceeds 0.6 under this model, the region is considered to have moderate rockburst tendency; if ω exceeds 0.8, it is considered that the region has strong rockburst tendency, which needs attention.

## 6. Discussion

Rock burst is a nonlinear dynamic process in which energy is accumulated in stable state and released in unstable state during the deformation and failure of a coal and rock mass system under specific geological conditions. It is a comprehensive reflection of the external environment, internal structure, and physical and mechanical properties. Its formation process is very complex, and the phenomenon has obvious spatiotemporal evolution characteristics [31]. So, the monitoring of rock burst is a complicated work. The past monitoring means focused more on a single monitoring means, such as microseismic monitoring [32] and stress monitoring, but the single monitoring method present difficulty in accurately measuring the trend of rockburst. So, in the current engineering application, multi-index fusion monitoring will be used to monitor the information of the coal mine and predict the possibility of rock burst, e.g., microseismic-stress joint monitoring, acoustic emission-electromagnetic radiation joint monitoring, and so on.

After collecting the microseismic data of the 1366 working face in Hengda coal mine, the time-frequency analysis method is used to analyze the spatiotemporal evolution information of coal and rock mass. Through the Fourier transform analysis of the microseismic signal, it can be seen that the overall amplitude of the high-energy event analyzed in this paper is large, and the main frequency is less than 100 Hz. This is obviously different from the microseismic events before the high-energy event, so it can be used as a discriminative index. After using another analysis method, namely wavelet packet transform [33], it can be seen that low-frequency components of the high-energy event account for more, and the features of the high-energy event can be further extracted. After adding the frequency band proportion of wavelet packet transform, the discrimination of the high-energy microseismic event will be more accurate. In addition, this paper uses the Hilbert–Huang transform [34], which can analyze the abrupt signal and achieve high accuracy in both time and frequency, so that the accuracy of capturing microseismic signal is higher. In this way, the first IMF component and the instantaneous energy are obtained.

The results obtained by the three time-frequency analysis methods are important judgment basis for distinguishing the high-energy event from events before the occurrence of high- energy event, which can be used as indicators for rock burst monitoring and warning. However, single microseismic monitoring struggles to fully reflect the changes in coal and rock mass, so it is necessary to introduce acoustoelectric monitoring into rock burst monitoring, which can improve the monitoring accuracy of rock burst. After obtaining the change characteristics of acoustoelectric indicators when the high-energy event occurs, this paper fuses the time-frequency analysis characteristics of microseismic monitoring and acoustoelectric characteristics with the analytic hierarchy process (AHP) to obtain the importance of each indicator, and then establishes the unascertained measure comprehensive evaluation model to monitor the risk of rock burst, so as to provide a timely warning.

The multi-parameter monitoring model established by the above analytic hierarchy process can accurately respond to both stable signal and burst signal. It can also have different sensitivity degrees to different indexes, which avoids the inaccurate judgment problem caused by the lack of quantification analysis in the past rock burst monitoring and warning. This method is also suitable for rock burst monitoring in different coal mine working faces.

The unascertained measure comprehensive evaluation model is adopted in this paper. How to build a more accurate and efficient model in rockburst monitoring is also the direction of future research. Nowadays, the support vector machine (SVM) algorithm [35], artificial neural network algorithm [36], and other computer algorithms are applied more and more frequently. This makes the application of machine learning in rock burst monitoring and warning very likely in the future [37]. This method allows for a more efficient and in-depth analysis of the data, but using this method requires a large amount of data input, so in-depth research is needed.

## 7. Conclusions

Through Fourier transform, wavelet packet transform, and Hilbert–Huang transform analysis of the representative high-energy event and the events before the high-energy event obtained from the 1366 working face in Hengda coal mine, the difference between the signal of the high-energy event and its precursor event is obtained. Through Fourier transform and wavelet packet transform joint analysis, it could be found that the main frequency of the high-energy event is low, but the amplitude of the main frequency is high, and the low frequency component occupies a relatively high proportion in the whole signal range, so the high-energy event belongs to the low frequency event. However, the main frequency and signal distribution of the events before the occurrence of the high-energy event are concentrated in the middle and high frequency bands. HHT verified these characteristics and further discovered the trend and characteristics of the high-energy event in instantaneous energy. The maximum instantaneous energy of the high-energy event is relatively high and shows an “inverted V type” trend that rises steeply from zero to the highest point and then drops steeply to zero. However, the events before the occurrence of the high-energy event show a fluctuation pattern in the whole sampling interval. Through the joint analysis of different time and frequency analysis methods, the discrimination limits of the high-energy event and the events before the occurrence of the high-energy event are obtained in the time and frequency domain.

Through the research on the recent acoustoelectric indicators of the high-energy event, it is found that when the high-energy coal and rock dynamic events occur, the acoustoelectric indicators show an obvious response. For example, there is a “rising type” trend that continues to strengthen and an “inverted V type” trend that falls after a significant increase to the maximum value in recent days. So, the criterion of acoustic emissions and electromagnetic radiation can be determined and used as the judgment basis for rockburst warning.

Based on the response rules of microseismic time-frequency characteristics and acoustoelectric indicators, the judgment index of the rockburst tendency of the 1366 working face of Hengda Coal Mine is obtained, including microseismic energy, main frequency, amplitude of main frequency, wavelet packet frequency band decomposition, the first IMF component of HHT, instantaneous energy, and acoustoelectric index.

The response range of each index corresponding to different grades of rockburst tendency was determined by the grading quantitative method. Then, a judgment matrix was established through the AHP to fuse the microseismic and acoustoelectric indicators and determine the weight of each indicator. Hence, the unascertained measure comprehensive evaluation model of the 1366 working face of Hengda Coal Mine was established.

## Figures and Tables

**Figure 1 ijerph-19-15698-f001:**
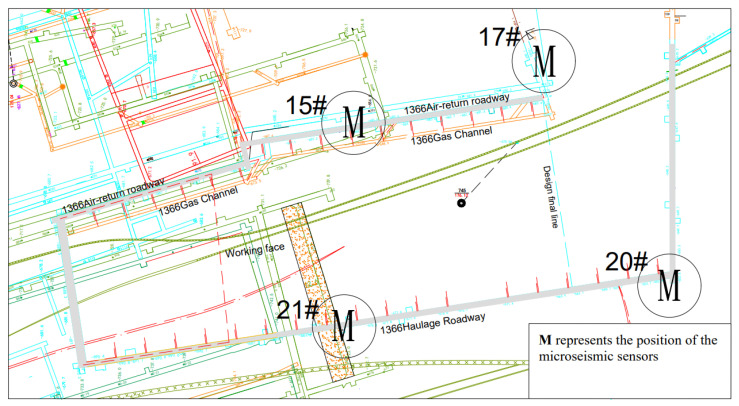
Layout of microseismic sensors in 1366 working face.

**Figure 2 ijerph-19-15698-f002:**
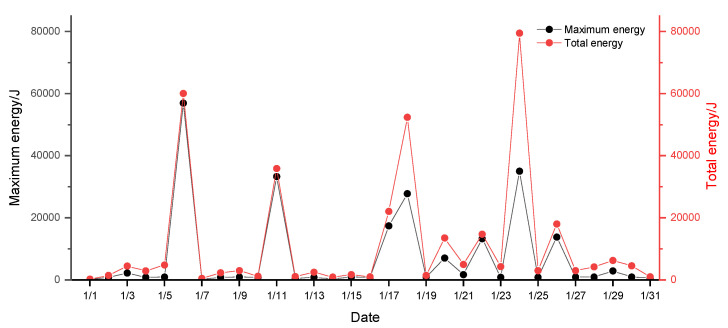
Sequential evolution of microseismic energy in January at 1366 working face of Hengda Coal Mine.

**Figure 3 ijerph-19-15698-f003:**
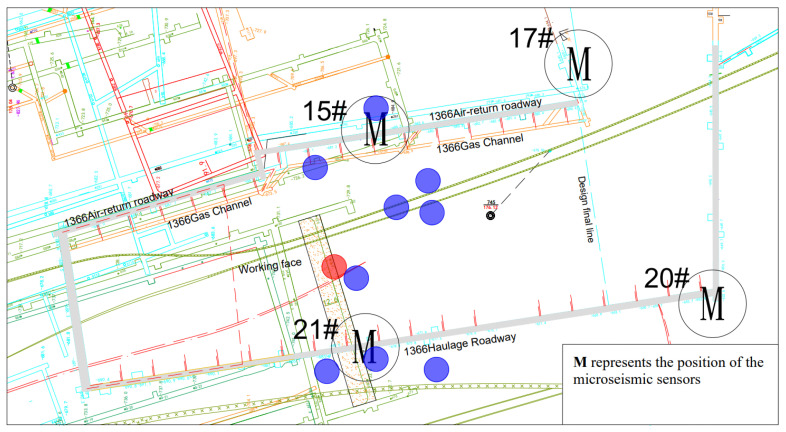
Distribution of microseismic events in 1366 working face.

**Figure 4 ijerph-19-15698-f004:**
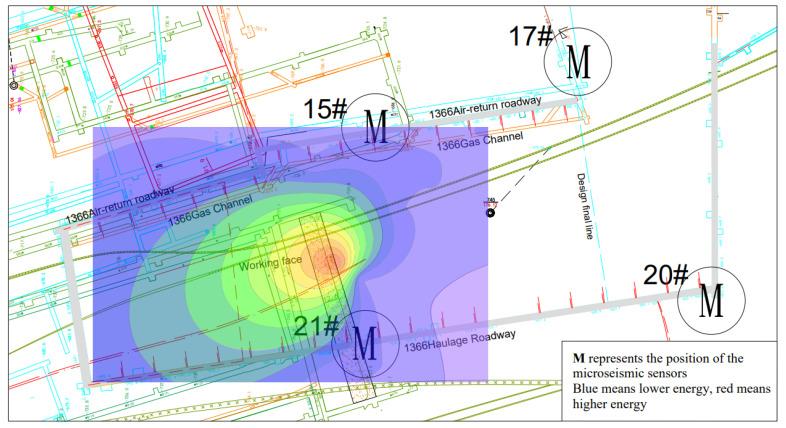
Cloud chart of microseismic energy of 1366 working face.

**Figure 5 ijerph-19-15698-f005:**
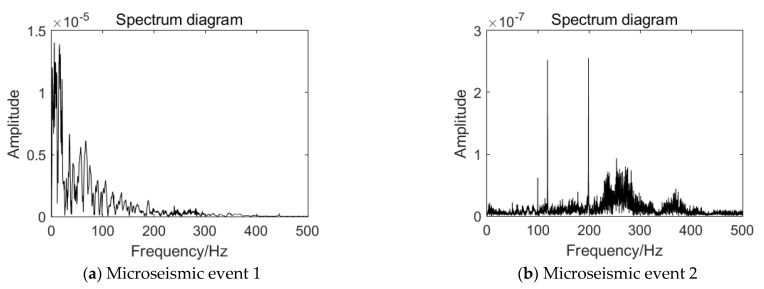
Waveform spectrum of microseismic events in working face.

**Figure 6 ijerph-19-15698-f006:**
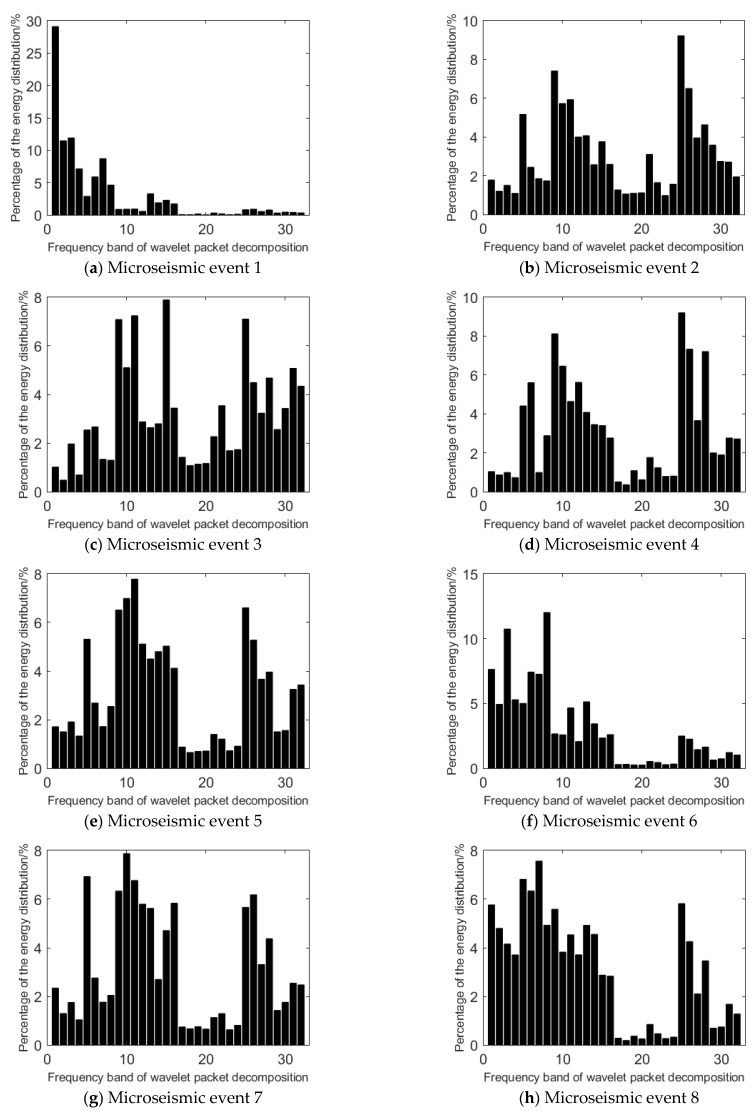
Frequency band energy distribution of microseismic event waveform.

**Figure 7 ijerph-19-15698-f007:**
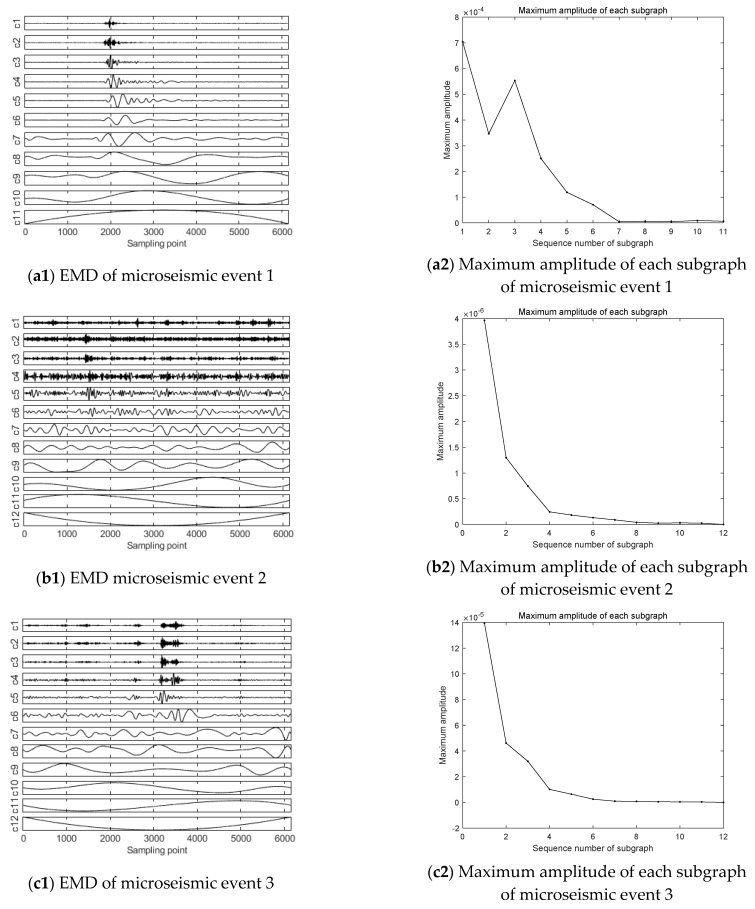
EMD and maximum amplitude of each subgraph.

**Figure 8 ijerph-19-15698-f008:**
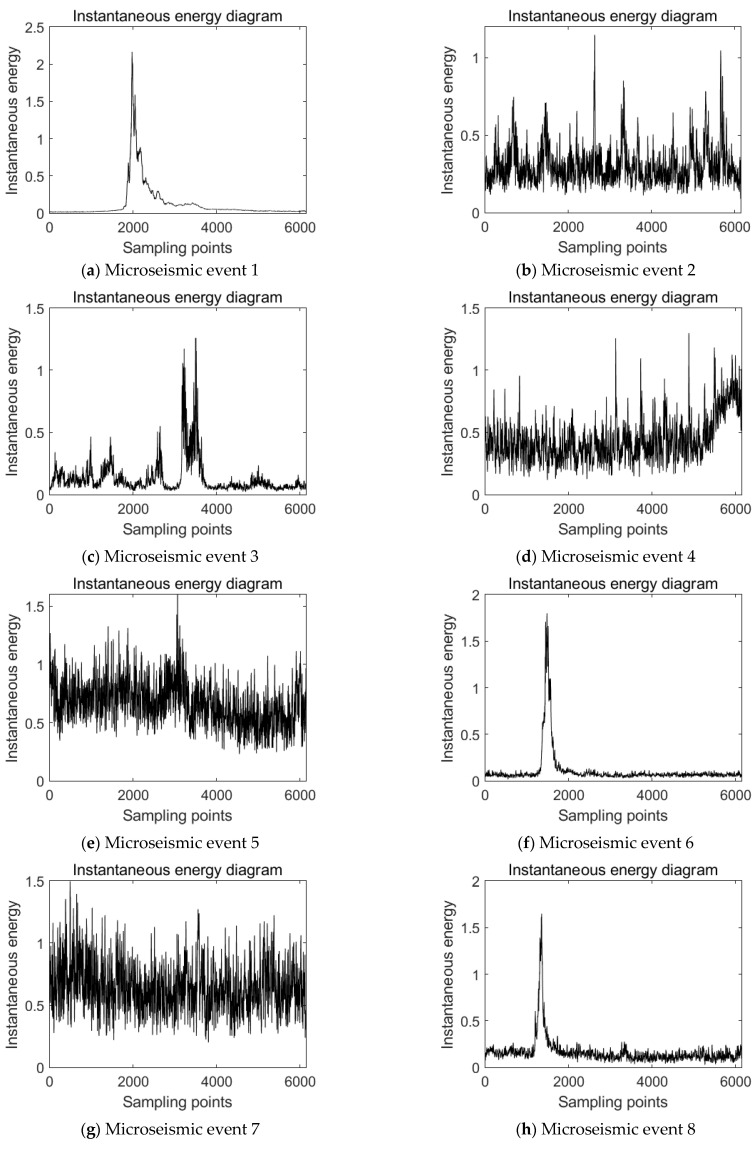
Instantaneous energy of microseismic events.

**Figure 9 ijerph-19-15698-f009:**
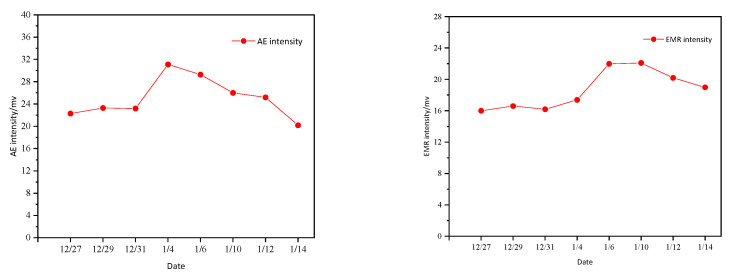
Acoustoelectric monitoring rules.

**Table 1 ijerph-19-15698-t001:** Basic information of microseismic events in working face.

Events	Date	Time	Location (x, y, z)/m	Energy/J
1	6 January 2022	12:39:09	x: 6397.88 y: 6541.51 z: −691.40	56,916.59
2	6 January 2022	11:29:08	x: 6184.52 y: 6626.60 z: −686.96	225.9
3	6 January 2022	4:26:41	x: 6433.59 y: 6453.45 z: −681.52	291.49
4	5 January 2022	20:00:46	x: 6388.26 y: 6439.05 z: −680.85	722.65
5	5 January 2022	11:56:05	x: 6418.16 y: 6528.30 z: −686.52	129.1
6	5 January 2022	11:17:42	x: 6376.65 y: 6629.09 z: −648.12	980.47
7	5 January 2022	9:59:54	x: 6303.50 y: 6642.51 z: −691.91	100.83
8	5 January 2022	3:19:45	x: 6528.15 y: 6660.61 z: −690.45	249.66
9	5 January 2022	1:56:18	x: 6454.82 y: 6591.46 z: −693.95	102.01
10	4 January 2022	17:55:40	x: 6481.62 y: 6611.86 z: −697.18	272.28

Note: x, y are horizontal coordinates, z is vertical coordinates.

**Table 2 ijerph-19-15698-t002:** Grading criteria for impact propensity evaluation indicators.

Evaluation Index		Classification	Scoring Criteria	Value Range
I_1_	Microseismic event energy	Level I (C_1_)	No rockburst tendency	≤1000
		Level II (C_2_)	Moderate rockburst tendency	1000~20,000
		Level III (C_3_)	Strong rockburst tendency	>20,000
I_2_	Dominant frequency	Level I (C_1_)	No rockburst tendency	>100
		Level II (C_2_)	Moderate rockburst tendency	50~100
		Level III (C_3_)	Strong rockburst tendency	≤50
I_3_	Dominant frequency amplitude	Level I (C_1_)	No rockburst tendency	≤10^−6^
		Level II (C_2_)	Moderate rockburst tendency	10^−6^~10^−5^
		Level III (C_3_)	Strong rockburst tendency	>10^−5^
I_4_	Wavelet packet decomposition (Divide the full frequency domain into 32 subbands)	Level I (C_1_)	No rockburst tendency	The energy distribution of 1~10 frequency bands accounts for less than or equal to 50%.
		Level II (C_2_)	Moderate rockburst tendency	The energy distribution of 1~10 frequency bands accounts for 50–70%.
		Level III (C_3_)	Strong rockburst tendency	The energy distribution of 1~10 frequency bands accounts for more than 70%.
I_5_	First IMF component of HHT	Level I (C_1_)	No rockburst tendency	≤10^−5^
		Level II (C_2_)	Moderate rockburst tendency	10^−5^~10^−4^
		Level III (C_3_)	Strong rockburst tendency	>10^−4^
I_6_	Instantaneous energy	Level I (C_1_)	No rockburst tendency	≤1.6
		Level II (C_2_)	Moderate rockburst tendency	1.6~2
		Level III (C_3_)	Strong rockburst tendency	>2
I_7_	Acoustoelectric index	Level I (C_1_)	No rockburst tendency	The intensity of AE and EMR is low, and there is no obvious change within one week.
		Level II (C_2_)	Moderate rockburst tendency	The intensity of AE and EMR is relatively high or has obvious fluctuation.
		Level III (C_3_)	Strong rockburst tendency	The intensity of AE and EMR has an obvious trend of continuous rise or strong fluctuation.

**Table 3 ijerph-19-15698-t003:** Judgment comparison scale criterion.

Scale	Definition and Description of the Comparison of Two Elements
1	Two elements are equally important (or equally strong)
3	One element is slightly more important (or slightly stronger) than another
5	One element is more important (or stronger) than another
7	One element is obviously more important (or obviously stronger) than another
9	One element is absolutely more important (or absolutely stronger) than another
2, 4, 6, 8	The scale of a compromise between the two scales

**Table 4 ijerph-19-15698-t004:** Calculation results of the weight of the comprehensive evaluation index of rock burst hazard.

Index I	Index-Weight α_i_
I_1_ Microseismic event energy	0.30
I_2_ Dominant frequency	0.20
I_3_ Dominant frequency amplitude	0.05
I_4_ Wavelet packet decomposition	0.10
I_5_ First IMF component of HHT	0.10
I_6_ Instantaneous energy	0.05
I_7_ Acoustoelectric index	0.20

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
