# Peer review of "Microseismic Dynamic Response and Multi-Source Warning during Rockburst Monitoring Based on Weight Decision Analysis"

_ijerph, 2022, doi:10.3390/ijerph192315698_

Round 1

Reviewer 1 Report

The manuscript presents interesting research, and the content is appropriate for the journal. Rockburst is a geological disaster induced by human in the construction of mineral resources mining and other projects, which is one of the major disasters in deep engineering. This study is of great significance to determine the warning indicators of rockburst, improve the accuracy of uncertainty quantitative analysis of rockburst and improve the discrimination accuracy of rockburst risk, based on the analysis of microseismic and the acoustoelectric index of 1366 working face in Hengda coal mine.

The manuscript is well written but there are still some questions, so I suggest “minor revisions” before publication.

1. The abstract needs to be further simplified and revised.

2. The introduction should be further summarize.

3. Add relevant legend in Figure 1 to make the description clearer.

4. Please further explain why these signal analysis methods are used.

5. Please further explain the advantages of monitoring and early warning based on weight analysis.

6. All variables in the formula should be declared or defined, please check them.

7. Some references cited are not accurate, please check them.

8. Some English expressions should be improved. Some sentences are too long and need to be broken down.

Author Response

Dear editors and reviewers:

Thank you for your letter and comments concerning our manuscript entitled “Microseismic dynamic response and multi-source warning during rockburst monitoring based on weight decision analysis” (Manuscript Number: IJERPH-1964906). Those comments are all valuable and very helpful for revising and improving our paper, as well as the important guiding significance to our researches. We have studied comments carefully and have made correction which we hope meet with approval. Revised portion are marked in red in the paper. The main corrections in the paper and the responds to the reviewer’s comments are as following:

Responds to the reviewer #1’s comments: The manuscript presents interesting research, and the content is appropriate for the journal. Rockburst is a geological disaster induced by human in the construction of mineral resources mining and other projects, which is one of the major disasters in deep engineering. This study is of great significance to determine the warning indicators of rockburst, improve the accuracy of uncertainty quantitative analysis of rockburst and improve the discrimination accuracy of rockburst risk, based on the analysis of microseismic and the acoustoelectric index of 1366 working face in Hengda coal mine.

The manuscript is well written but there are still some questions, so I suggest “minor revisions” before publication.

  1. The abstract needs to be further simplified and revised.

Response 1: Thank you very much for your important comment. The abstract has been further simplified and revised. Please see the attachment.

  1. The introduction should be further summarize.

Response 2: Thank you very much for your important comment. The introduction has been further summarized. Please see the attachment.

  1. Add relevant legend in Figure 1 to make the description clearer.

Response 3: Thank you very much for your important comment. Please see the attachment.

  1. Please further explain why these signal analysis methods are used.

Response 4: Thank you very much for your important comment. As for the reasons for using these signal analysis methods, I mainly supplemented them in chapters 3.2 and 3.4.  Please see the attachment.

  1. Please further explain the advantages of monitoring and early warning based on weight analysis.

Response 5: Thank you very much for your important comment. The advantages of monitoring and early warning based on weight analysis are supplemented in introduction and section 5. Please see the attachment.

  1. All variables in the formula should be declared or defined, please check them.

Response 6: Thank you very much for your important comment. The formulas in the paper are supplemented. Please see the attachment.

  1. Some references cited are not accurate, please check them.

Response 7: Thank you very much for your important comment. The references have been revised.  Please see the attachment.

  1. Some English expressions should be improved. Some sentences are too long and need to be broken down.

Response 8: Thank you very much for your important comment. I have made some improvements to the English expressions in some parts of the text. Please see the attachment.

We appreciate for Editors/Reviewers’ warm work earnestly, and hope that all these changes fulfil the requirements to make the manuscript acceptable for publication in IJERPH.

Once again, thank you very much for your comments and suggestions.

If you have any questions, please feel free to contact me.

Sincerely Yours

Reviewer 2 Report

Reviewer Comments

Paper title: Microseismic dynamic response and multi-source warning during rockburst monitoring based on weight decision analysis

This paper applies three time-frequency analysis methods of Fourier Transform, wavelet packet transform and Hilbert Huang transform to extract the characteristics of microseismic signals, and then combines acoustoelectric monitoring to carry out multi-source warning, so as to determine the classification standard of impact tendency evaluation indicators, and then uses AHP to determine the weight of each index, and establishes the unascertained measure comprehensive evaluation model in 1366 working face of Hengda coal mine, it can be a good reference for evaluating the safety of rockburst warning and realizing the risk management in the monitoring process.

A manuscript has a practical application and also provides important theoretical for the next studies.

The paper can be accepted for publication after providing the corrections mentioned below.

Point 1. The abstract section sounds unclear. It seams to be as a part of the conclusion section.

You should rephrase text mentioned below by decreasing the total length.

The analysis shows that:(1)compared with the events before high-energy event, the main frequency of high-energy event is lower, and the amplitude is larger in the low-frequency band. The overall energy of high-energy event is mainly concentrated in the low-frequency band, with obvious low-frequency characteristics.(2)The highest instantaneous energy is higher when high energy event occurs, while the instantaneous energy tends to zero during the rest of sampling time, showing a "inverted V" trend of increasing steeply from zero to the highest instantaneous energy and then gradually decreasing to zero.(3)When rock burst occurs, the acoustoelectric index shows a trend of "rising" or "inverted V" or other strong changes”.

Point 2. Keywords need to be modified. Please use words not combinations of words or phrases.

Point 3. In the Introduction section, an enhanced literature review is required. It will be great if the authors show some description in context – Why it is important to conduct this study? Can the expected result be used or implemented within other coal basins? If yes, then how? What limitations?

Point 4. References are placed in the text in a wrong format. The numeration must be given [1], [2]-[5], etc.

Point 5. The aim and the tasks must be highlighted at the end of the Introduction section.

Point 6. It is quite difficult to read the paper. Why do authors not prepare the paper using a commonly known IMRaD structure?

The study should follow a conventional pattern. Methods, Results and Discussion need to be mentioned clearly so that the readers can easily understand the gist of the manuscript.

Point 7. What do you think about the possibility of the usage of machine learning and deep learning application in mine microseismic event classification? Can it be implemented in your issue. Please consider the reference mentioned below in your paper.

Jinqiang, W., Basnet, P., & Mahtab, S. (2021). Review of machine learning and deep learning application in mine microseismic event classification. Mining of Mineral Deposits, 15(1), 19-26. https://doi.org/10.33271/mining15.01.019

Point 8. In general, the presented article leaves a positive impression and, after eliminating these comments and taking into account the recommendations made, it can be recommended for publication in the journal "IJERPH".

Author Response

Dear editors and reviewers:

Thank you for your letter and comments concerning our manuscript entitled “Microseismic dynamic response and multi-source warning during rockburst monitoring based on weight decision analysis” (Manuscript Number: IJERPH-1964906). Those comments are all valuable and very helpful for revising and improving our paper, as well as the important guiding significance to our researches. We have studied comments carefully and have made correction which we hope meet with approval. Revised portion are marked in red in the paper. The main corrections in the paper and the responds to the reviewer’s comments are as following:

Responds to the reviewer #2’s comments: This paper applies three time-frequency analysis methods of Fourier Transform, wavelet packet transform and Hilbert Huang transform to extract the characteristics of microseismic signals, and then combines acoustoelectric monitoring to carry out multi-source warning, so as to determine the classification standard of impact tendency evaluation indicators, and then uses AHP to determine the weight of each index, and establishes the unascertained measure comprehensive evaluation model in 1366 working face of Hengda coal mine, it can be a good reference for evaluating the safety of rockburst warning and realizing the risk management in the monitoring process.

A manuscript has a practical application and also provides important theoretical for the next studies.

The paper can be accepted for publication after providing the corrections mentioned below.

Point 1:The abstract section sounds unclear. It seams to be as a part of the conclusion section.

You should rephrase text mentioned below by decreasing the total length.

“The analysis shows that:(1)compared with the events before high-energy event, the main frequency of high-energy event is lower, and the amplitude is larger in the low-frequency band. The overall energy of high-energy event is mainly concentrated in the low-frequency band, with obvious low-frequency characteristics.(2)The highest instantaneous energy is higher when high energy event occurs, while the instantaneous energy tends to zero during the rest of sampling time, showing a "inverted V" trend of increasing steeply from zero to the highest instantaneous energy and then gradually decreasing to zero.(3)When rock burst occurs, the acoustoelectric index shows a trend of "rising" or "inverted V" or other strong changes”.

Response 1:Thank you very much for your important comment. The abstract has been modified. Please see the attachment.

Point 2: Keywords need to be modified. Please use words not combinations of words or phrases.

Response 2: Thank you very much for your important comment. Please see the attachment.

Point 3: In the Introduction section, an enhanced literature review is required. It will be great if the authors show some description in context – Why it is important to conduct this study? Can the expected result be used or implemented within other coal basins? If yes, then how? What limitations?

Response 3: Thank you very much for your important comment. Please see the attachment.

Point 4: References are placed in the text in a wrong format. The numeration must be given [1], [2]-[5], etc.

Response 4:Thank you very much for your careful comment. The citation format and references have been revised. Please see the attachment.

Point 5: The aim and the tasks must be highlighted at the end of the Introduction section.

Response 5: Thank you very much for your careful comment. Please see the attachment.

Point 6: It is quite difficult to read the paper. Why do authors not prepare the paper using a commonly known IMRaD structure?

The study should follow a conventional pattern. Methods, Results and Discussion need to be mentioned clearly so that the readers can easily understand the gist of the manuscript.

Response 6: Thank you very much for your suggestion. Under your advice, I have made appropriate adjustments to the structure of this paper. The Methods in this paper is divided into the following parts: The time-frequency analysis method is located in Chapter 3. Chapter 4 introduces the analysis of the acoustoelectric characteristics, which is necessary for the introduction of the multi-parameter research method in Chapter 5. Chapter 5 introduces the analytic hierarchy process and the establishment method of the unascertained measure comprehensive evaluation model. In addition, I supplement the Discussion in Chapter 6. Chapter 7 is the content of Conclusion. Please see the attachment.

Point 7: What do you think about the possibility of the usage of machine learning and deep learning application in mine microseismic event classification? Can it be implemented in your issue. Please consider the reference mentioned below in your paper.

Jinqiang, W., Basnet, P., & Mahtab, S. (2021). Review of machine learning and deep learning application in mine microseismic event classification. Mining of Mineral Deposits, 15(1), 19-26. https://doi.org/10.33271/mining15.01.019

Response 7: Thank you very much for your important comment. The application of machine learning in rock burst monitoring and warning is very possible. For example, the study of multi-parameter monitoring method in this paper uses analytic hierarchy process to establish the unascertained measure comprehensive evaluation model, but I believe that the machine learning can also be used in multi-parameter fusion research. This approach allows for a more efficient and in-depth analysis of the data, but using this approach requires a large amount of data input, so further analysis is needed. 

Point 8: In general, the presented article leaves a positive impression and, after eliminating these comments and taking into account the recommendations made, it can be recommended for publication in the journal "IJERPH".

Response 8: Thank you very much for your important comment. Please see the attachment.

We appreciate for Editors/Reviewers’ warm work earnestly, and hope that all these changes fulfil the requirements to make the manuscript acceptable for publication in IJERPH.

Once again, thank you very much for your comments and suggestions.

If you have any questions, please feel free to contact me.

Sincerely Yours

Round 2

Reviewer 2 Report

Dear authors, I am satisfied with the corrections provided by you.

But, as for my previous suggestion mentioned below:

Point 7: What do you think about the possibility of the usage of machine learning and deep learning application in mine microseismic event classification? Can it be implemented in your issue. Please consider the reference mentioned below in your paper.

Jinqiang, W., Basnet, P., & Mahtab, S. (2021). Review of machine learning and deep learning application in mine microseismic event classification. Mining of Mineral Deposits, 15(1), 19-26. https://doi.org/10.33271/mining15.01.019

You give me a response mentioned below:

Response 7: Thank you very much for your important comment. The application of machine learning in rock burst monitoring and warning is very possible. For example, the study of multi-parameter monitoring method in this paper uses analytic hierarchy process to establish the unascertained measure comprehensive evaluation model, but I believe that the machine learning can also be used in multi-parameter fusion research. This approach allows for a more efficient and in-depth analysis of the data, but using this approach requires a large amount of data input, so further analysis is needed. 

At the same time, you should indicate this information in your paper not only in response to the reviewer. Your colleagues from the University of Science and Technology Beijing and Tongji University published good work that can and should be considered in your paper and add it to the reference list.

Author Response

Dear editors and reviewers:

Thank you for your letter and comments concerning our manuscript entitled “Microseismic dynamic response and multi-source warning during rockburst monitoring based on weight decision analysis” (Manuscript Number: IJERPH-1964906). Those comments are all valuable and very helpful for revising and improving our paper, as well as the important guiding significance to our researches. We have studied comments carefully and have made correction which we hope meet with approval. Revised portion are marked in red in the paper. The main corrections in the paper and the responds to the reviewer’s comments are as following:

Point 7: What do you think about the possibility of the usage of machine learning and deep learning application in mine microseismic event classification? Can it be implemented in your issue. Please consider the reference mentioned below in your paper.

Jinqiang, W., Basnet, P., & Mahtab, S. (2021). Review of machine learning and deep learning application in mine microseismic event classification. Mining of Mineral Deposits, 15(1), 19-26. https://doi.org/10.33271/mining15.01.019

You give me a response mentioned below:

Response 7: Thank you very much for your important comment. The application of machine learning in rock burst monitoring and warning is very possible. For example, the study of multi-parameter monitoring method in this paper uses analytic hierarchy process to establish the unascertained measure comprehensive evaluation model, but I believe that the machine learning can also be used in multi-parameter fusion research. This approach allows for a more efficient and in-depth analysis of the data, but using this approach requires a large amount of data input, so further analysis is needed. 

At the same time, you should indicate this information in your paper not only in response to the reviewer. Your colleagues from the University of Science and Technology Beijing and Tongji University published good work that can and should be considered in your paper and add it to the reference list.

Response 7: Thank you very much for your important comment. I have indicated this information in my paper, and add it to the reference list. Please see the attachment.

We appreciate for Editors/Reviewers’ warm work earnestly, and hope that all these changes fulfil the requirements to make the manuscript acceptable for publication in IJERPH.

Once again, thank you very much for your comments and suggestions.

If you have any questions, please feel free to contact me.

Sincerely Yours
